# Causes and implications of the unforeseen 2016 extreme yield loss in the breadbasket of France

Tamara Ben-Ari[1], Julien Boé [2], Philippe Ciais[3], Remi Lecerf[4], Marijn Van der Velde[4] & David Makowski[1]

In 2016, France, one of the leading wheat-producing and wheat-exporting regions in the world suffered its most extreme yield loss in over half a century. Yet, yield forecasting systems failed to anticipate this event. We show that this unprecedented event is a new type of compound extreme with a conjunction of abnormally warm temperatures in late autumn and abnormally wet conditions in the following spring. A binomial logistic regression accounting for fall and spring conditions is able to capture key yield loss events since 1959. Based on climate projections, we show that the conditions that led to the 2016 wheat yield loss are projected to become more frequent in the future. The increased likelihood of such compound extreme events poses a challenge: farming systems and yield forecasting systems, which often support them, must adapt.

[1] INRA, AgroParisTech, Université Paris-Saclay, UMR 211 Agronomie, Thiverval-Grignon 78850, France. [2] CECI, Université de Toulouse, CERFACS/CNRS, Toulouse 31100, France. [3] Laboratoire des Sciences du Climat et de l'Environnement, 91191 Gif-sur-Yvette, France. [4] European Commission, Joint Research Centre (JRC), Via E. Fermi 2749, Ispra, VA 21027, Italy. Correspondence and requests for materials should be addressed to T.B-A. (email: tamara.ben-ari@inra.fr)

Crop yield forecasting systems rely on a combination of expert knowledge, data mining and analysis, and mechanistic and/or statistical models[1–3]. The increased extreme weather conditions in recent years[4], have challenged these systems. Heat waves and drought events take the largest toll on production[5], and are anticipated to increase in frequency and severity in the northern mid-latitudes[6]. Extreme yield losses also can occur, however, due to events resulting from insidious constellations of climate variables forming a compound extreme. The 2016 extreme loss of wheat harvest in the breadbasket region of France is one such example, and we present here an in-depth analysis of this event and its implications for wheat yield forecasting.

France ranks fifth in the global league table of national wheat production, and despite its limited arable area, produces more wheat than any other country in the European Union (EU). This is achieved because of very high yields. Recent French yields equal about 7.4 t ha$^{-1}$. In comparison, China, India, Russia and the United States, the world's four largest wheat producers, harvest about 5, 2.5, and 3 t ha$^{-1}$, respectively[7]. Between 2000 and 2013, France was the EU's main grain exporter, exporting about 17 million tonnes of wheat mainly to North Africa, where local production covers only 10–50% of the demand[7].

The 2016 winter-wheat harvest was disastrous. Yields in the breadbasket region dropped on average by 27.7% compared to trend expectations (Fig. 1) and by 39.5% compared to 2015. This equates to a shortfall of about 8 million tonnes compared to the 24.5 million tonnes usually harvested in this region or of about 11 million tonnes compared to the record 27.5 tonnes harvested in 2015[8]. These extremely low yields combined with lower exchange prices on international markets compared to 2015 induced a substantial income loss for farmers and about 2.3 billion dollars loss for France's trade balance[9].

None of the public forecasting systems anticipated the magnitude of this loss. Even just before the disastrous harvest, forecasts predicted average yields of 7.23 t ha$^{-1}$, close to the 5-year average[10], overestimating the actual value by about 2 t ha$^{-1}$. Towards the end of the growing season, there were concerns among regional experts about heavy rainfalls leading to flooding

and saturated agricultural soils in the Seine river basin, and about high incidences of foliar diseases. High observed wheat biomass at the very end of winter on the one hand and a strong confidence on the effectiveness of fungicides on the other, possibly explain that close-to-average yield values were anticipated by most experts until harvest started unfolding in mid-July. Although technical experts have since investigated the possible causes of this extreme yield loss, no quantitative study has characterized the precise climatic conditions that led to it—we still have little understanding of why yield forecasts failed by such a large amount in 2016. Here, we analyse long-term yield and climate time series at the scale of departments (administrative units) from 1959 to 2016 and address the following research questions: (i) how exceptional were climate conditions, individually and combined, in the breadbasket region during the 2015–2016 growing season? (ii) Can 2016 help us improve forecasting systems? And (iii) will such events become more frequent in the future?

We first search for single and compound climatic extremes that occurred during the 2015–2016 growing season. We then perform a statistical analysis to model severe and extreme wheat yield losses since 1959 based upon these climate variables. We show that the huge wheat yield loss in 2015–2016 can be predicted from a combination of climate variables related to higher temperature in autumn and wetter conditions in spring. Finally, based on future climate warming projections, we show that the specific conditions that led to the 2016 wheat yield loss are projected to become more frequent towards the end of the century.

## Results

**Extreme loss and unprecedented weather conditions.** We focus this study on the breadbasket region, comprising 27 departments, and which together accounts for more than 67% of France's total wheat production (average since 1959). In 2015, these departments for example produced more wheat than all of Ukraine and slightly less than all of Canada. All 27 departments suffered extreme yield losses in 2016, leading to the spatio-temporal

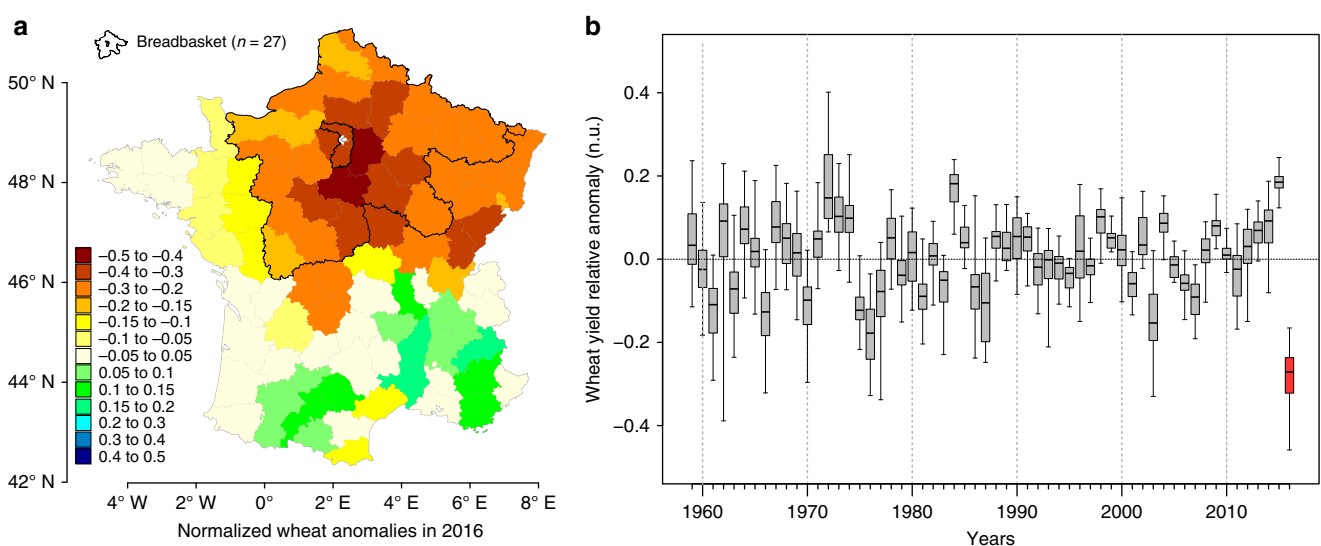

**Fig. 1** Spatio-temporal pattern of the 2016 extreme yield loss. **a** Wheat yield anomalies in 2016 relative to expected values defined in each department by the long-term yield trend (e.g., −0.1 corresponds to a loss of 10% compared to expectation, see Methods). The breadbasket region of France is delineated in bold black contours. Note that a similar figure is presented for each year in the data set in the Supplementary Movie 1. The map was generated with R based on the yield data used in the analyses. **b** Boxplot of the distribution of anomalies in the breadbasket (1959–2016). Decades are separated with thin dotted lines. Anomalies in 2016 are highlighted in red. Yearly median anomalies are presented in Supplementary Fig. 1b

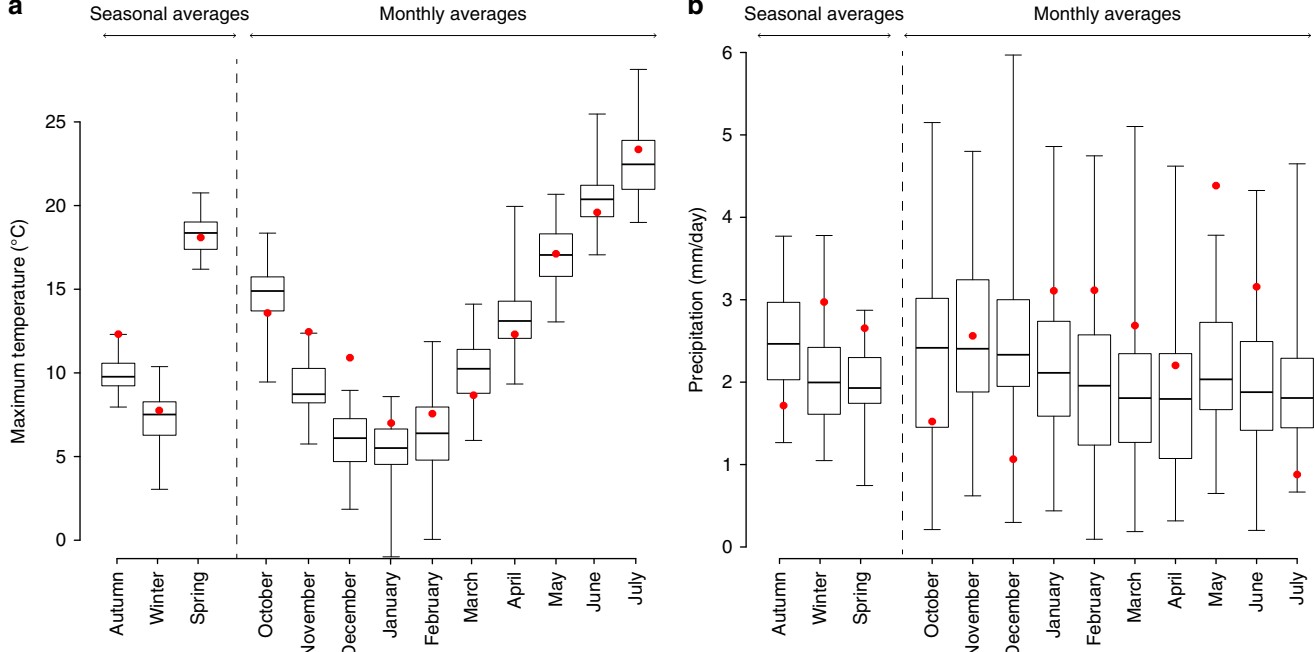

**Fig. 2** Maximum temperatures and precipitation over the 1959–2016 wheat growing seasons. Boxplot of **a** daily maximum temperature and **b** daily precipitation averaged each year over the study area for the period 1959–2015. Whiskers extend to maximum and minimum values. Values in the 2015–2016 growing season are presented as red dots (for other climate variables see Supplementary Fig. 2)

pattern shown in Fig. 1 and Supplementary Fig. 1. Before 2016, yields never dropped by more than 15% of the long-term mean, on average over the study area, except during the extreme drought of 1976 when the loss was about 16%. By contrast, in 2016, yields were between 17 and 45% below their expected values. The year 2016 thus suffered the single most extreme loss experienced over the past five decades (Fig. 1b, Supplementary Fig. 1b, Supplementary Movie 1).

To characterize climate conditions in the breadbasket region, we present monthly and seasonal distributions for seven growing-season climate variables from the autumn of 1958 to the 2016 harvest using data from the SAFRAN analysis[11]. The 2015–2016 growing season is singled out and indicated by red dots (Fig. 2, Supplementary Fig. 2). Both maximum and minimum temperatures in the late autumn (here November and December) of 2015 were exceptionally high (Fig. 2a, Supplementary Fig. 2a), resulting in a dramatic reduction of the number of vernalizing days (see Supplementary Fig. 2b). Vernalization is a critical process that controls the development of wheat through exposure to cold temperatures. Temperatures were also high in January and February of 2016 (i.e., close to the third quartile) and, overall, the winter of 2016 was relatively wet compared to the average over 1959–2015. The data in Fig. 2 also show that the spring (here April to July inclusive) of 2016 was extremely wet, with mean precipitation of 2.66 mm day$^{-1}$, conditions associated with abnormally low potential evapotranspiration (in particular in May and June see Supplementary Fig. 2c). We found only three other years with whole spring conditions within 10% of spring 2016 values (i.e., 1983, 2000, and 2012). Note that seasonal values hide between-months variability. In April 2016, precipitation was close to average, but in May it reached a record high of 4.39 mm day$^{-1}$, which had never occurred since 1959. June 2016, the month preceding harvest, was also characterized by the persistence of high precipitation (Fig. 2b, Supplementary Fig. 2d). Daily average solar radiation was low during most of the spring in 2016, with a record low value of 160 W m$^{-2}$ in June 2016 (Supplementary Fig. 2e).

Overall, the 2015–2016 growing season was characterized by a unique combination of abnormally warm temperatures in the late autumn and abnormally high precipitation, with concurrent low radiation and potential evapotranspiration, in the spring. These variables were outside the 95th percentile of their distributions. The joint probability of all the 2015–2016 growing season climate conditions was null, which makes this event a compound extreme.

**Relating yield loss to autumn and spring conditions**. We designed an ensemble of statistical models to diagnose yield loss occurrence as a function of time series of climate variables in each department since autumn 1958. We considered probabilities of yield losses below −10 and −15% (respectively corresponding to a loss below one standard deviation and below the 10th percentile). In the rest of the manuscript we refer to these levels as severe and extreme yield losses and present results for net yield losses (i.e., negative yield anomalies) in the supplement.

The influential parsimonious subset of climate variables was selected using the Bayesian Information Criterion (BIC) independently for each yield loss level. We considered both monthly and seasonal climate variables with autumn defined as October–December, winter as January–March and spring as April–July, inclusive. We first computed BIC for each variable independently and select a subset of four best covariates relying on the extreme climate events characterizing the 2015–2016 growing season. Climate variables in the selected three best models are consistent across the levels of loss considered. These are: (1) the number of days with maximum temperatures between 0 and 10 °C in December, (2) precipitation in November, (3) precipitation in spring (or in May), and (4) minimum temperature in June. Interactions among these variables are considered (see full description of each model in Supplementary Table 2). The same variables are selected when the models are fitted to normalized anomalies (See Supplementary discussion and Supplementary Fig. 3).

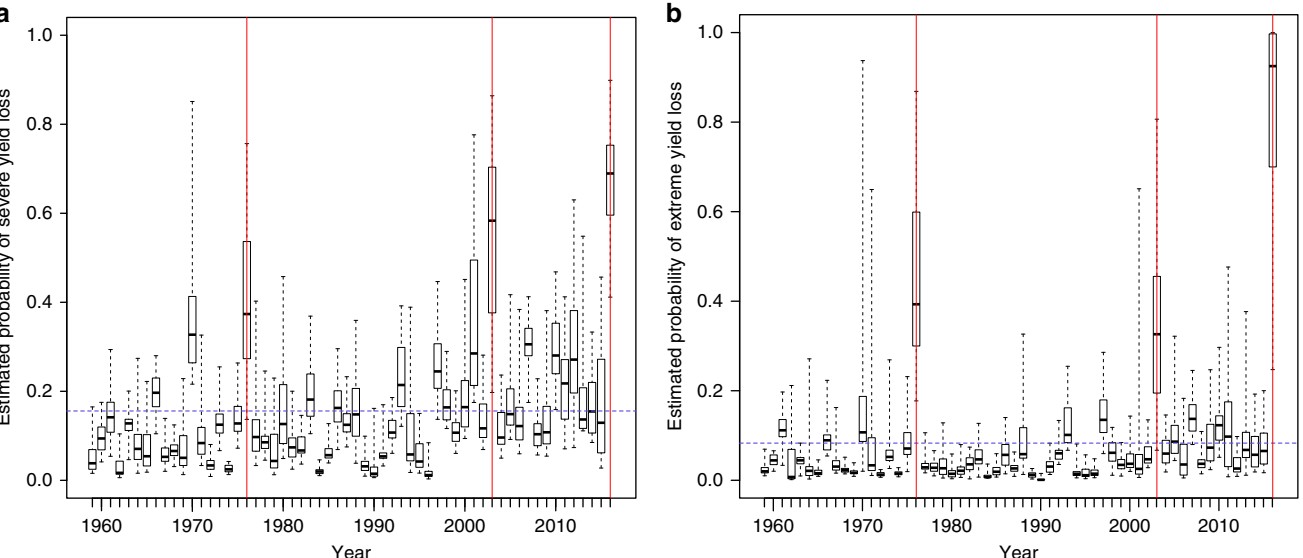

**Fig. 3** Modeled probabilities. Estimated probabilities of **a** severe and **b** extreme yield loss (yields more than 10 and 15% lower than the expected value) from generalized linear regression models based on climate predictors fitted to the full yield time series (1959–2016) in the breadbasket region. The dotted blue line is the prior probability (the empirical proportion of severe or extreme yield loss events in the dataset). As an indication, vertical red lines correspond to median yield loss occurrences across all departments larger than −15% (i.e., in 1976, 2003, and 2016). Note that the figure for net yield losses is presented in the Supplementary Fig. 3

The influential climate variables statistically associated with yield losses over 1959–2016 include those identified by the French academy of agriculture, extension services or specialized journalists after the 2016 harvest (summarized in Supplementary Table 1). The selected best models, combining the four-abovementioned variables and between-seasons interactions, trained on the entire data set, are used to assess the risk of net, severe, and extreme yield loss in the breadbasket over all harvest years (Fig. 3, Supplementary Fig. 4).

The models successfully capture a series of key yield-loss events, e.g., extreme losses in 1976, 2003, and 2016 and severe losses in 1966, 1970, and 2007. As an example, the estimated probability of extreme yield loss reaches a median (10th–90th percentiles) across departments of 0.39 (0.21–0.74) during the severe drought year of 1976, the second worst yield on record and of 0.33 (0.15–0.59) for the heatwaves of 2003 (Fig. 3). Note that probabilities computed by our statistical models need to be interpreted as a departure from prior probabilities defined as the frequency of actual losses in the data set (Supplementary Table 3). For example, the prior value for extreme losses is 0.081, computed probabilities thus correspond to an increase of roughly 5 and 4-folds in 1976 and 2003 respectively. Notably, the years 1976, 2003, and 2016 are highly contrasted in terms of autumn and spring conditions (Supplementary Fig. 5).

We find that the selected statistical models all include an interaction between the number of days between 0 and 10 °C in December and spring precipitation ($p = 0.0027$, Supplementary Table 2) suggesting that the strength of the relationship between high precipitation in the spring and the probability of yield loss increases with temperatures in the late autumn. For example, according to our model, if the number of days between 0 and 10 °C in December drops from 20 to 10 and the following spring is characterized by average precipitation levels, the probability of severe yield loss increases from 0.12 to 0.2. But when precipitation in the following spring is above one standard deviation this probability increases from 0.2 to 0.51 (Fig. 4a). In other words, the effect of excess precipitation in the spring is stronger if it follows warmth in late autumn. Such an effect is also observed for extreme (Fig. 4b) or net losses (Supplementary

Fig. 6). Selected models also include an interaction between temperatures in June (i.e., flowering period) and precipitation over the spring for all yield loss levels (see discussion section).

Our results illustrates the importance of taking into account the joint occurrence of multiple yet specific climate predictors during the growing season[12] if we are to successfully predict the impacts of compounded extremes like the one of 2016. Although we did not attempt to use our models for neighboring countries, we note that similar weather anomalies in autumn and spring were recorded in Belgium, which also suffered an extreme yield loss of about 24%[13]. Our results could thus probably be extrapolated to other similar agro-ecological areas, provided wheat-cropping systems use similar varieties and agricultural practices.

**On the prediction of the 2016 extreme yield loss.** Figure 5 presents odds ratios computed from estimated probabilities of severe and extreme yield loss over each unit of the breadbasket from logistic models trained in the study area on a dataset excluding 2016 (out-of-sample procedure). Odds ratios indicate the relative chance of severe or extreme yield losses. According to our statistical models, the odds of an extreme yield loss in 2016 were 35 times higher than expected (i.e., from prior values, Supplementary Table 3). This is equivalent to a risk ratio of about 11. In other words, our statistical models estimates a probability of extreme yield loss 11 times higher than a priori expected. Our models also predict between 1.8 and 4.6 more chance of losing yield severely in 2016 than in an average year. Estimated probabilities of severe and extreme yield loss in 2016—based on two separate models excluding 2016 from the training datasets—are on average of 0.46 and 0.71, respectively (Supplementary Table 3). Note that these values hide important local disparities (Supplementary Fig. 7); both the confidence intervals and the inter-unit ranges are larger for extreme yield losses reflecting the smaller number of occurrences of such events in our data set. The results obtained are robust to a change in the training and test data sets (see Methods section).

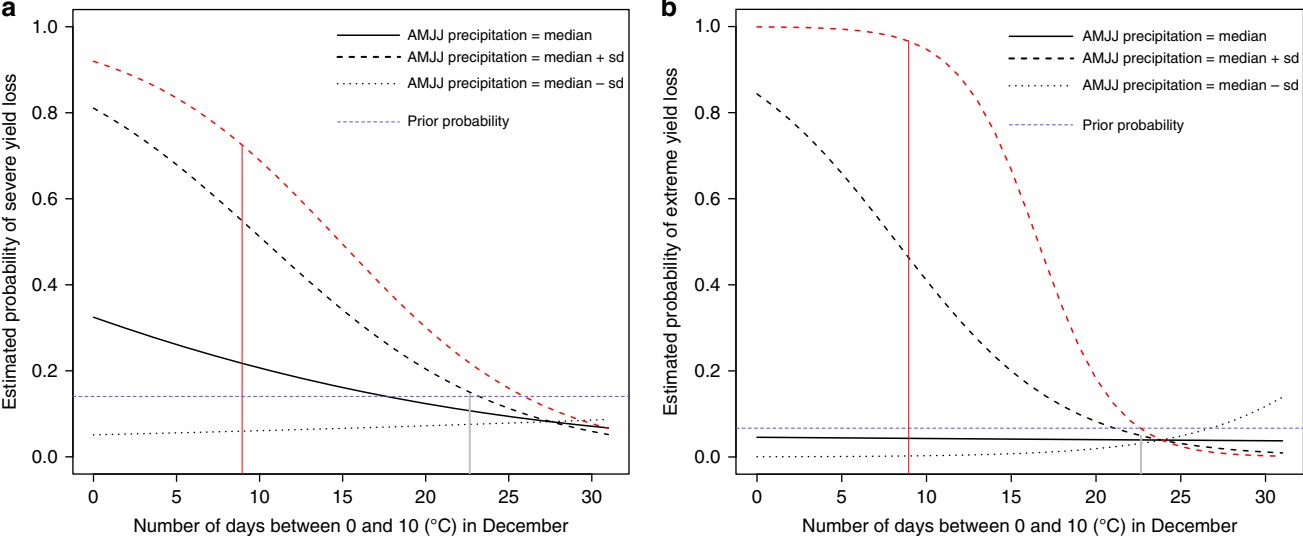

**Fig. 4** Autumn–spring interaction. Modulation by spring precipitation of the relationship between the number of days between 0 and 10 °C in December and the probability of severe (**a**) and extreme wheat yield loss (**b**). Interactions are shown here for normal precipitation (daily AMJJ precipitation equal to the median over 1959–2016; bold line), and for wet and dry spring conditions (daily AMJJ precipitation equal to +/− one standard deviation; dotted lines). Vertical grey line: median number of days with maximum temperature between 0 and 10 °C over 1958–2016; red dotted lines: modeled value for 2016

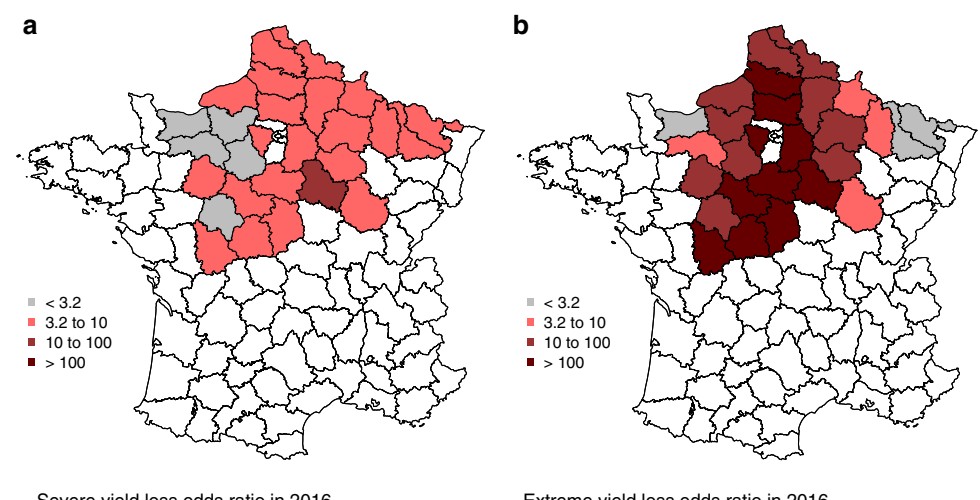

Severe yield loss odds ratio in 2016          Extreme yield loss odds ratio in 2016

**Fig. 5** Odds of yield loss in 2016. Odds ratios of severe (**a**) and extreme (**b**) yield loss in the study area from the best models trained on yield data excluding 2016. Data are shown in Supplementary Table 3. Models include temperature and precipitation conditions in the late autumn and spring. We rely on the odds ratio classification by[68] with values from 1 to 3.2 not worth more than a bare mention, between 3.2 and 10 suggesting substantial evidence, from 10 to 100 strong and above 100 decisive. The maps were generated with R based on the yield data used in the analyses

Thus, providing that all weather data were available in time, the use of our statistical model before the 2016 harvest would have indicated strong evidence of a severe to extreme yield loss. Unlike an actual forecast, however, the influential set of climate covariates were selected from a data set including the 2015–2016 growing season. The implication of this result lies in the key information provided by 2016 extreme wheat yield loss: outlining the necessity of considering late-autumn climatic conditions for winter wheat forecasting.

**What 2016 presages for the future**. Based on climate projections from the Coupled Model Intercomparison Project phase 5 (CMIP5), we assessed how regional climate change will impact the likelihood of climate conditions similar to the ones experienced during the 2015–2016 wheat growing season. All four

climate variables identified as influential were extracted from model outputs. Very small changes in spring and November or spring precipitation over northern France are projected under the RCP2.6 scenario, i.e., the frequency of 2016-like precipitation anomalies hardly changes over the twenty-first century (Fig. 6a, b). On the other hand, as a result of the warming trend under the RCP2.6 scenario, the number of days with Tmax between 0 and 10 °C in December is projected to decline by on average 5 days (around 22%) by the end of the twenty-first century compared to the 1950s (Fig. 6c). The likelihood of a temperature anomaly at least as severe as during the 2015–2016 growing season is extremely small in the 1950s (~2%), slightly greater in the current decade (~6%), and increases moderately by the end of the twenty-first century (~12%). The ensemble-mean June minimum temperature is projected to increase by 1.5 °C during the twenty-first century under the RCP2.6 scenario (Fig. 6d). The relative

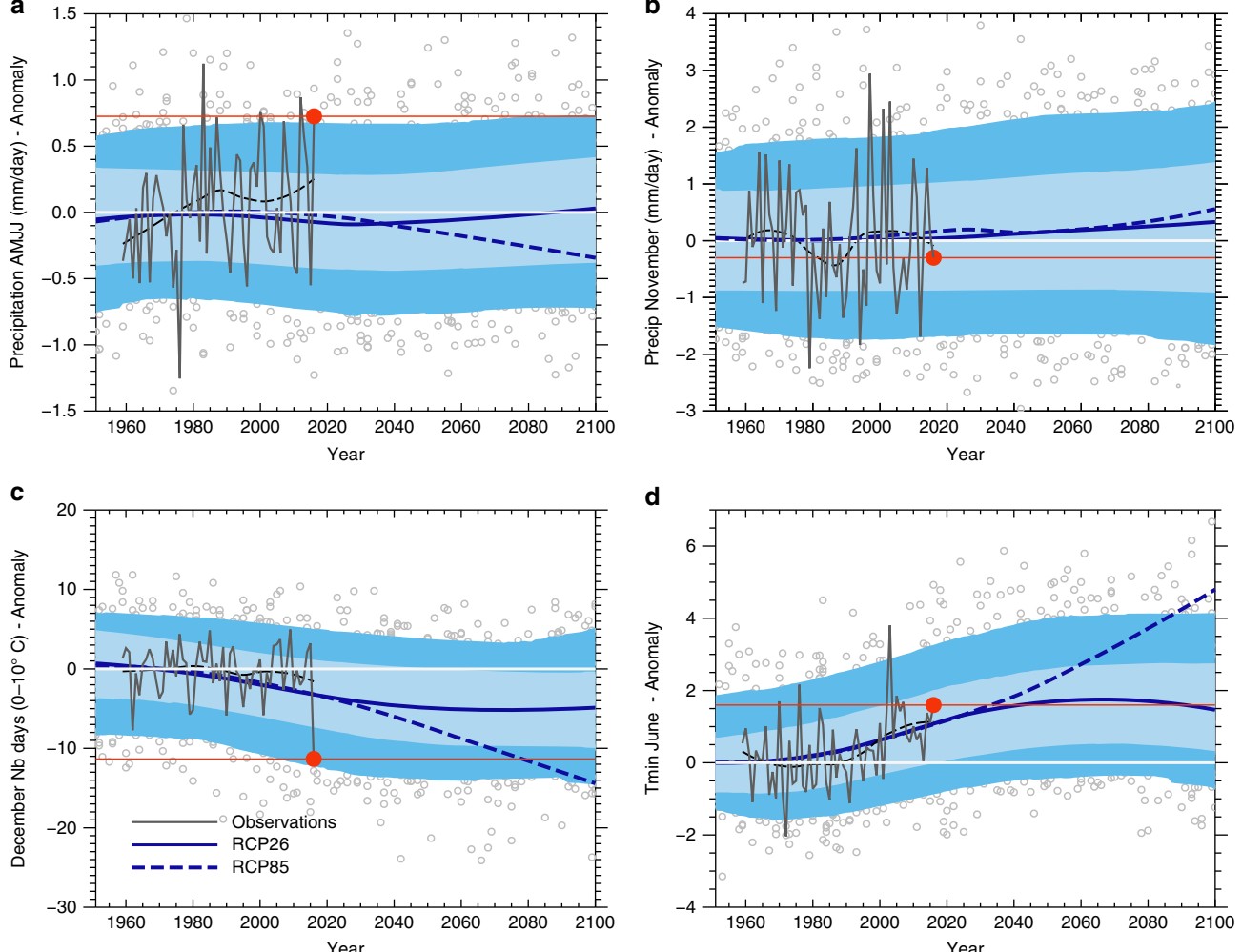

**Fig. 6** Time evolution towards 2100 of the four climate variables identified as key in our statistical models. **a** Precipitation in spring (mm per day), **b** Precipitation in November (mm per day), **c** Number of days with Tmax between 0 and 10 °C and **d** Minimum temperature in June (°C). The anomalies relative to the 1959–1988 reference period are shown in Supplementary Fig. 8. The blue lines show the multi-model signal derived from an ensemble of 13 CMIP5 models and estimated by locally weighted scatterplot smoothing (loess) (solid line: RCP2.6 scenario. Dashed line: RCP8.5 scenario). SAFRAN reanalysis data are shown with dark grey lines (thick dashed line: loess results, thin line: yearly values). The light (dark) blue shaded areas show the 20–80% (5–95%) range of the model outputs distribution assuming the independence of models, for the historical and RCP26 simulations. The associated delimiting quantiles are estimated with a local polynomial quantile regression. The grey points show the yearly model outputs

cumulative frequency of the positive June 2016 minimum temperature anomaly in the CMIP5 ensemble decreases from around 90% in 1951–1980 to 70% in the 30-year period centered on 2016 and 50% at the end of the twenty-first century.

Despite the near-absence of projected change in springtime precipitation (here including July), given the strong projected twenty-first century increase in warm minimum temperature anomalies in June and the decrease in the number of days between 0 and 10 °C in December, climate conditions favorable to yield loss are projected to become more frequent under the RCP2.6 scenario. Our results also suggest that the warming trend observed in France over the last several decades—and partly attributable to anthropogenic forcings[14,15]—have already increased the probability of 2016-like climate conditions occurring.

The potential benefits of the aggressive mitigation policies required to follow a low warming scenario like RCP2.6 are obvious when the previous results are compared to the ones obtained with the RCP8.5 scenario (dashed blue lines in Fig. 6—see inter-model spread in Supplementary Fig. 8). In this intensive

warming scenario, a number of vernalizing days as small as in December 2015 becomes the norm by 2070 and, even if associated with a slight drying of spring months, would drastically increase the probability of a 2016-like event. Also, June 2016 minimum temperature, unusually warm relatively to the mid-twentieth century climatology, would become characteristic of an extremely cold June by the end of the twenty-first century (Supplementary Fig. 8c).

Note that we did not address the possible effects on wheat yields of an increase in atmospheric $CO_2$ concentration. $CO_2$ effects are expected to manifest through increased leaf-level carboxylation rates and stomatal closure. Both processes interact with each other, and stomatal closure has the dual consequence of saving soil water and increasing surface temperature by reducing transpiration. It is thus expected that elevated $CO_2$ would improve yields under dry condition[16,17].

To summarize, there are two key climatic factors associated with the 2016 loss: autumn temperature for which we know with great confidence that 2016 is going to move closer to an average year and spring precipitation for which we do not detect a

noticeable time trend. Qualitatively, 2016-like years are very likely to become more frequent in the future since one of the two extreme factors will no longer be extreme anymore. However, a robust quantification of future yield loss probability from a 2016-like growing season is beyond the scope of our study and possibly hampered by the high uncertainty associated with projected changes in precipitation.

## Discussion

The experts who have been analysing the possible cause of the extreme yield loss, all recognized a posteriori the abnormal precipitation and radiation conditions in the spring of 2016 and, some of them, the warmth of the preceding late autumn. Cited mechanisms most often include lodging, and more widespread occurrence of pests and diseases (Supplementary Table 1).

A decrease in the number of days between 0 and 10 °C in December (i.e., vernalizing days[18])—corresponding to warmth in a usually cold month—may have direct and/or indirect effects on wheat yield. The lack of sufficient vernalizing days can modify subsequent phenological development[18]. Vernalization for example affects the number of leaves and tillers, floral initiation time or flowering phenology[19]. Warmth during the vernalization period can delay the onset of the reproductive stage increasing the risk of exposure to high temperatures during anthesis[20].This effect alone is unlikely, however, to explain the loss observed, at least in 2016, because most soft wheat varieties cultivated in France require only about 40 vernalizing days[21], and this requirement was still fulfilled by the end of the winter, despite warm temperatures (Supplementary Fig. 2b). This is corroborated by the independent observation that spring varieties—with no vernalization requirement—were also strongly affected in 2016 (i.e., about −20% at the national scale[8]). But, winter warmth is also known to shift the phenology of pests and diseases, causing earlier colonization of crops and earlier spreading of vector-borne viruses[22–24] resulting in more frequent and severe infections[25]. In addition, positive precipitation anomalies or persistent moisture facilitates the development and spread of fungal diseases in the spring[25]. A plausible hypothesis to our findings is that the combination of a mild autumn/winter favors a build-up of parasites and a persistence of inoculum, which subsequently leads to large-scale disease prevalence in the field provided conducive spring precipitation conditions would occur[26]. 2016 was marked by very high to abnormal precipitation levels in the spring; conditions indeed favorable for the spread of diseases. Moreover, localized extreme precipitation events—a phenomenon observed in the field in 2016—may also induce flooding which subsequently leads to anoxia and lodging[27,28].

The 2016 wheat harvest assessment revealed lower grain numbers and very low grain weight, suggesting impaired grain filling[29]. Minimum temperatures around June determine the length of the grain-filling period, with higher temperatures lowering kernel weights[30,31]. In 2016, the minimum temperatures in June were abnormally high (Supplementary Fig. 2a). In our models, minimum June temperature indeed appears to be an important co-factor influencing yield loss. Additionally, we find a significant interaction between June temperatures and spring precipitation: the latter modulate the probability of yield loss from June heat. We find that increased spring precipitation can slightly downplay the importance of high minimum temperatures in June in the models whereas dry conditions increase loss probability from high temperatures in June (e.g., for net yield losses in Supplementary Fig. 6). This interaction more generally reflects the impact of heat and water stress on photosynthetic activity. Finally, and consistent with earlier analyses for wheat and maize yield[32,33], the best statistical models also found a small

positive relationship between November precipitation (i.e., between sowing and emergence) and yield loss. This could suggest a negative effect of waterlogging on root growth[34] and/or point to enhanced survival or growth of soil-borne diseases[35]. Note that November 2015 precipitation was close to the 50-year average (Fig. 2b).

The failure of yield forecasts in 2016 needs to be understood in the context of a difficulty in simulating winter crops compared to spring crops[36] yields. This is perhaps because of the wide range of growth drivers and limiting factors in wheat[33,37], whereas maize yields are for example more evidently driven by water and heat stresses[38]. An additional deficiency of deterministic wheat forecasts is the difficulty in simulating development stages in coincidence with climate events. A common example is the impact of heat stress during[39–41] or after[42] anthesis. Complex and localized phenomena such as flooding, lodging, or the prevalence of pests and diseases, which can take a large toll on production, are ignored in both process-based and statistical models[43]. Spatially-explicit reliable information of initial soil water conditions, rooting depth or soil drainage and soil water-holding capacity[44–47] should be included in assessing the risk-benefit balance of wet years such as 2016. Models could also benefit from region-specific parameterization of agro-management practices or onset-adaptation strategies. To overcome the shortcomings of using a single crop model, the use of model ensembles is arguably the way forward[48,49], but this strategy is not yet routine in seasonal crop forecasting and would imply implementing the above-mentioned mechanisms. Finally, there is readily available information that could be harnessed to improve forecast systems. A regional plant health bulletin, for example, is published each month in France[50], and hydrological anomalies are regularly updated[51]. These observational data could complement forecast estimates by improving an analyst's judgment. Turning available information from local sources into harmonized data sets at regional scale, updated on an operational basis should probably be a key priority to improve wheat yield forecasts in Europe. Early-warning procedures making real-time information available to farmers in exchange for targeted field observations could also help improve forecasts (e.g., early yield estimates collected during harvest through social media[52]).

Other abnormal climate conditions have affected primary production in the past. These most often occurred during droughts: for example, the 2003 heatwave, which caused tremendous damage to vegetation in France[53], had an enormous societal impact[54]. This was also true in 1976, a growing season with similar limiting factors. Those visible impacts of water and heat stresses in the spring may actually have hidden other multivariate climatic events with similar or higher impacts on wheat harvests (e.g., in 1970, 1987, 2007, or 2016).

Based on long-term wheat yield and the department-scale climate time series, we show that the compound interaction between temperature in the late autumn/early winter and precipitation in the spring is the key to understanding the severe yield drop of 2016. A series of red flags were identified that could have enabled experts to anticipate the event. Our results also show that probabilistic approaches can be very helpful for anticipating yield losses provided that they are properly trained on the right combination of climate variables. Depending on the nature of decision makers' demand, crop yield analysts may consider combining a deterministic approach with probabilistic analyses.

Wheat growing in the breadbasket region is overwhelmingly composed of highly-mechanized wheat monocultures heavily relying on the use of fertilizers and pesticides[55]. Despite the steady use of fungicides[56] over the last decade, and intensified use of chemical inputs in 2016, a widespread disease occurrence severely impacted wheat yields. There are ecological arguments

suggesting that monocultures are less resilient to abnormal climate events[57] and more sensitive to disease outbreaks than more complex cropping systems or landscapes[58–60]. On the other hand, is the call for a loosening of government restrictions on the use of herbicides and pesticides in order to deal with yield fluctuations[61]. These opposing recommendations will not only shape future wheat production in France but also its transformative path toward climate change adaptation.

## Methods

**Yield and climate data**. We analysed winter-wheat yield time series in France for the period 1958–2016 at the spatial scale of departments (administrative units known in French as départements) based on official survey data[8]. In each department we applied a detrending method to crop-yield data to remove the long-term effect of technological improvements within the study period. Relative yield anomalies $\bar{a}_{i,t}$ are defined as:

$$\bar{a}_{i,t} = \frac{(Y_{i,t} - \mu_{i,t})}{\mu_{i,t}},\qquad(1)$$

where $Y_{i,t}$ is the yield value and $\mu_{i,t}$ the expected yield value in the $i$th unit at year $t$. Expected yield values, $\mu_{i,t}$ are estimated using local regressions (loess). We define severe and extreme yield loss from relative anomalies below −10 and −15% of expected yields. We also define net yield loss as having all negative relative anomalies.

Climate conditions are spatially uniform at the scale of these administrative units (typically 30–100 km across), thus ensuring coherence between climate data sets and the impacted wheat yield. The studied area is also relatively homogenous in terms of wheat production systems[55] topography or weather conditions. No crop mask was used because initial tests showed no difference in climate between the entire territory of administrative units and their cropland-covered fraction (not shown). The winter-wheat growing season starts with sowing in October, undergoes a vernalizing period in winter, and ends at harvest in the following July. We refer to the growing season using harvest years (e.g., 2016 encompasses October 2015 to July 2016). Our data set covers October 1958 to July 2016 for harvests occurring from 1959 to 2016. Input climate data are from the SAFRAN reanalysis[11,62] updated by the French weather service from October 1958 to July 2016. The data cover France on an $8 \times 8$ km grid on a daily time step. We computed monthly values in each grid cell for the following variables: average maximum (Tmax) and minimum temperatures (Tmin) (°C), average precipitation (mm d$^{-1}$), average solar radiation (W m$^{-2}$), average Penman-Monteith potential evapotranspiration (mm d$^{-1}$), number of days with Tmax between 0 and 10 °C (i.e., vernalizing days[18]) average number of rainy days with precipitation per month, from October to July (the wheat growing season). Monthly data were also averaged during October–November–December (OND) and April–May–June–July (AMJJ) henceforth called autumn and spring. Climate data were then aggregated over the territory of each administrative unit.

We calculated the frequency of occurrence of climate conditions more extreme than those of the 2015–2016 growing season as follows. Let $X_{i,t}$ be the value of a climate variable (monthly or seasonal mean) during the $t$th growing season (1959–2015) in the $i$th administrative unit. $X_{i,2016}$ is the value of the same variable during the 2015–2016 growing season

$$I_{i,t} = 0 \quad \text{if } X_{i,t} \leq X_{i,2016}$$

$$I_{i,t} = 1 \quad \text{if } X_{i,t} > X_{i,2016}$$

The frequency of a value $X$ strictly superior to the one of the 2015–2016 growing season is given by

$$\text{Freq}_i = \frac{\sum_{t=1}^{N} I_{i,t}}{N},\qquad(2)$$

where $N$ is the number of growing seasons ($N = 58$). We then averaged $\text{Freq}_i$ over all administrative units ($n = 27$) within the study area into $\overline{\text{Freq}}_i$ and identified extreme regional variables during the 2015–2016 growing season as those with $\overline{\text{Freq}}_i < 0.05$ or $\overline{\text{Freq}}_i > 0.95$ (i.e., the average of occurrence frequencies of $X$ across all the administrative units is lower than 0.05 or higher than 0.95).

The number of years with maximum (alt. minimum) temperatures in December exceeding the value of December 2015 is null in all of the 27 administrative units. November was also extremely warm (Fig. 2). Temperatures during the autumn of 2015 have a frequency of occurrence of 0–0.05. For May precipitation, May potential evapotranspiration and radiation in June, the frequency of years exceeding the value of 2016 is also below 5%, Similarly, no more than 5% of the years have a minimum temperature over the growing season higher than that of 2016.

Considering the conjunction of October–November–December temperatures and April–May–June–July (spring) precipitation averaged over the study area, the

year 2016 is a single outlier, in conditions opposed to the ones of the drought year 1976 (the second most important yield drop on the time period considered, Supplementary Fig. 5). On average over the study area, the closest years to 2016 for autumn temperatures and spring rainfall are 2007 followed by 2012, 2001, and 1995. The years 1987, 2013, and 2001 were those with the highest number of administrative units close to 2016 for of spring precipitation. The year 1995 was the closest to 2016 for the total (low) number of vernalizing days. Years 2012, 2007, and 1995 were the closest to 2016 for autumn maximum temperatures.

**Statistical analyses**. We used a suite of binomial logit regression models trained using climate and yield data to estimate the probability of net, severe and extreme yield loss from climate inputs. Both monthly and seasonal averages of climate data were used to construct different models. Among all possible models including a single input variable, we selected the most parsimonious ones, according to their Bayesian Information Criteria (BIC), to identify the most influential predictor climate variable. We then used a stepwise selection procedure to identify the best combinations of input variables, with and without interactions. The model with the lowest BIC was finally selected for each level of yield loss independently (i.e., three best models, one per level of yield loss, Supplementary Table 2). During the model selection process, each model was fitted by maximum likelihood to binary data indicating occurrence of net, severe and extreme yield losses (see SI). Computations were done with the functions glm (family = binomial), predict.glm, and step.glm of R (Version 3.1.0).

The variables found to be influential for predicting yields are consistent over all three best models (Supplementary Table 2). These variables are:

- The number of days with Tmax between 0 and 10 °C in December
- November precipitation or average number of rainy days in November
- Minimum June temperature
- AMJJ or June precipitation.

Two interactions are also selected, namely between the number of days between 0 and 10 °C and AMJJ precipitation, and between minimum June temperatures and AMJJ precipitation. The estimated parameter values of the selected models are presented in Supplementary Table 2. Probabilities of yield loss in 1959–2016 are computed in each of the 27 units of the breadbasket using each selected model. To test the robustness of our results to the definition of the study area, we computed the probability of loss over a larger number of administrative units (including those outside of the breadbasket, i.e., 35 and 45 administrative units, see Supplementary Fig. 7) with the model trained on the study area. To predict the occurrence of yield loss in 2016, the selected models were fitted to the time series excluding 2016, i.e., from 1959 to 2015, and the predictive probabilities of yield loss in 2016 were computed as above. Results for severe and extreme yield losses are presented in Fig. 3 and Supplementary Fig. 4 for net losses.

Reported probabilities must be interpreted as a departure from a prior probability with this probability set to the frequency of yield losses in the samples. The odds obtained with the prior probabilities and with the statistical model are defined by:

$$O_{\text{prior}} = \frac{P_{\text{prior}}}{1 - P_{\text{prior}}} \text{ and } O_{\text{stat}} = \frac{P_{\text{stat}}}{1 - P_{\text{stat}}}\qquad(3)$$

with $P_{\text{prior}}$ equal to 0.47 (alt. 0.46, 2016 excluded) for net yield loss; 0.16 (alt. 0.14, 2016 excluded) for severe yield loss, and 0.083 (alt., 0.077 2016 excluded) for extreme yield loss. The probabilities $P_{\text{stat}}$ are the ones computed from the statistical model (with or without the 2016 data). The odds ratio thus corresponds to the ratio of $O_{\text{stat}}$ to $O_{\text{prior}}$. We also refer to the risk ratio $r$, defined by

$$r = \frac{P_{\text{stat}}}{P_{\text{prior}}}\qquad(4)$$

**Climate projections**. Climate projections from the Coupled Model Inter-comparison Project phase 5 (CMIP5)[63] were used for the 1951-2100 time period. Before 2005, we used the so-called historical simulations, in which the climate models are forced by the historical evolution of the main natural and anthropogenic forcings. After 2005, the results of two Representative Concentration Pathways (RCP) scenarios[64] are contrasted.

The RCP2.6 scenario (RCP26) assumes aggressive mitigations policies to likely limit global warming since the pre-industrial period to 2 °C[65]. This scenario is therefore close to the objectives of the Paris Agreement on Climate. The RCP8.5 scenario (RCP85) assumes a "business as usual" approach to the climate change issue, and results in a global warming close to 4 °C at the end of the 21st century[66].

Climate projections from a subset of 13 models (bcc-csm1-1-m, BNU-ESM, CanESM2, CCSM4, CNRM-CM5, CSIRO-Mk3-6-0, GFDL-CM3, HadGEM2-ES, IPSL-CM5A-MR, MIROC5, MPI-ESM-MR, MRI-CGCM3, and NorESM1-M) were analyzed. Only the models with both the RCP26 and RCP85 scenarios, and all the variables necessary for our study have been selected. We also only used one climate model by modeling center, to limit the lack of independence within the ensemble, given the strong similarity that generally exists between same center

models[67]. We then assume the independence of the selected models, as most studies to date and the IPCC report[66].

Different climate variables identified as influential for yield loss are extracted from raw model outputs on their native grids. Then, the spatial averages are computed on a domain that encompasses the 27 units of interest and whose boundaries are: 45.5°N, 51.5°N, −1.5°E, 8°E. Only the grid points with a fraction of land greater than 75% are used. The same domain is used to compute the spatial averages for SAFRAN climate indices.

Anomalies relative the 1959–1988 reference period are analyzed to deal with the potential mean climatological biases in the CMIP5 projections. We then make the implicit hypothesis that the model distributions for the inter-annual anomalies are realistic. This assumption is reasonable given the purpose of our analyses and is not crucial for our conclusions. For instance, the relative cumulative frequencies of the 2016 anomalies on the 1959–2016 period are close in the models and SAFRAN (96.5%, 85%, 40%, 3% in the models versus 93%, 90%, 43%, 2% in SAFRAN for, respectively, spring precipitation, June minimum temperature, November precipitation and the number of vernalizing days in December).

**Data availability**. The yield data that support the findings of this study are available from the corresponding author upon reasonable request. SAFRAN data are accessible here: http://mistrals.sedoo.fr/HyMeX/Data-Access-Registration/?project_name=HyMeX. CMIP data are accessible here: https://esgf-node.llnl.gov/projects/esgf-llnl/

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

## Acknowledgements

This work was supported by the CLAND convergence institute funded by the French National Research Agency (ANR). T.B.-A. and D.M. acknowledge the INRA-CIRAD meta-program GloFoods (PrevSaison). P.C. acknowledges the support of the European Research Council Synergy grant ERC-2013-SyG-610028 IMBALANCE-P. The authors thank Météo France (Direction de la Climatologie et des Services Climatiques) for providing the SAFRAN data set. We acknowledge the World Climate Research Program's Working Group on Coupled Modeling, which is responsible for CMIP, and we thank the climate modeling groups that developed the models listed in the method section of this paper for producing and making available their model output. For CMIP the U.S. Department of Energy's Program for Climate Model Diagnosis and Inter-comparison provides coordinating support and led development of software infra-structure in partnership with the Global Organization for Earth System Science Portals. The authors would also like to thank Phil Tajitsu Nash for editing, Emmanuelle Gour-dain for helping us retrieve some of the wheat data, and Marie Launay, Antoine Gar-darin, Gilles Grandeau and Dominique Le Floch for insightful discussions.

## Author contributions

T.B.-A., J.B., P.C. and D.M. framed the study. T.B.-A. and J.B. completed the data sets. T.B.-A. and D.M. performed statistical analyses of historical data. J.B. analysed CMIP5 data. All authors interpreted the results. J.B., R.L. & M.V.d.V. contributed to the writing of specific sections of the manuscript; T.B.-A., P.C. and D.M. wrote the paper.
