## [Peer Review File · Nature Communications]

Reviewers' comments:

Reviewer #1 (Remarks to the Author):

I found this study to be very interesting. It takes a recent event that is puzzling, asks how unusual it was in the historical context, and develops an understanding of its causes and spells out implications for the future. I learned quite a bit from it. I think it should be published, but I have a few questions/suggestions.

My main question is about the model itself. The authors make an unconventional choice in modeling the occurrence of loss above certain amounts (i.e. a binary variable) rather than modeling the yield itself. This raises the question of whether conclusions would be much different if modeling yield itself, or %yield anomaly. I would suggest the authors conduct this alternative analysis and present the results. They could either revisit variable selection for this new outcome variable, and/or could test predictions if using the same predictors but a different outcome. This would help strengthen the paper and convince the reader the exact specification isn't too important (or if it turns out it is, the authors should explain why their approach makes more sense)

The paragraph starting on L49 could use some more details on the yield loss, particularly what was associated with it in terms of diseases. This becomes an important part of the story but is only really clarified on L347 that there was a specific disease that was thought to cause much of the loss.

L155-156. Not clear why the second sentence says from 0.61 and not from 0.66

Figure legends and axes are hard to read. Please make font bigger when possible

My understanding was that severe losses are anything > 10% whereas extreme is >15%. Therefore severe has to be more likely than extreme, by definition. Yet on L233-34 it says "Estimated probabilities of net, severe, and extreme yield loss in 2016 are on average of 0.71, 0.46 and 0.8 respectively"

L382 (reference to wrong table)

I'd suggest a simple figure at the end showing the probability of extreme yield loss over time for rcp2.6 and 8.5. this would aid discussion already in the text, and provide a nice summary visual for the main point. If space is needed, fig 5 could be moved to supplement.

The supplemental figures were not numbered nor did they have captions in the version I have, which is annoying.

Reviewer #2 (Remarks to the Author):

GENERAL COMMENTS

The manuscript NCOMMS-17-18455 by Ben-Ari et al., entitled "Causes and implications of the unforeseen 2016 extreme yield loss in Europe's breadbasket" tackles a very important, hot research Topic, i.e. crop yield impacts of extremes, and addresses some fundamental research questions, such as: (1) how exceptional were the climate conditions, individually and combined (2) why did the existing yield forecasting Systems fail to anticipated the extreme yield loss? (3) how can - on the basis of this experience and detailed analyses - wheat yield forecasting systems be improved in the future. (4) will the observed combined events become more frequent in the future?

The research presented is original, yields several novel results and may have considerable influence on future directions of crop modelling /yield forecasting. The paper presents a sound combination of knowledge and methods in the fields of climate projections, crop modelling and yield forecasting. Moreover, the paper is very well written, contains key references and uses adequate figures (very nice illustrating key results) with very few exceptions (see, specific comments). Very minor points found that ask for amendment (see, specific comments)

Overall judgement: minor revision

SPECIFIC COMMENTS

Page 2,

line 51: you may want to mention the total production achieved in 2015

line 53-55 would be interesting to know how much the low exchange prices contributed to income loss in comparison to the yield reduction effect

Page 6,

Lines 117-119: please, add a brief explanation on why you choose the values below 0, -10 and -15% for net, severe and extreme? – some reference to earlier /other work?

Lines 121-122 maybe you could explain why you did categorically not consider climate variables of months January, February and March?

Lines 128-129 could you say a bit more (than summary in Table S1) about the composition of expert panels and the method of expert interviews? – would be interesting to learn more about this

Page 9,

Lines 174-178 I would suggest to also refer here (or later in the discussion) to a related paper on the expected higher frequency of adverse weather on wheat production in Europe ->

Trnka, M., Rötter, R.P., Ruiz-Ramos, M. Kersebaum, K-C., Olesen, J.E. & Semenov, M.A. (2014).

Adverse weather conditions for wheat production in Europe will become more frequent with climate change. *Nature Climate Change*, 4, 637-643 (doi: 10.1038/nclimate2242).

Lines 174-178 inform how the lack of sufficient vernalizing days can modify subsequent development

Lines 192-193 – would be good if you could find quantitative information on how much durum wheat was affected (and possibly a reference?)

Lines 197 to 201 (page 10) – is there some literature already that supports your hypotheses? - please, check for such and add here

Page 10,

Lines 202-210 this section, though interesting, does not say much about the exact mechanisms leading to yield loss –specify the prevailing processes and their effects, please

Lines 211-215 – this becomes now a bit too speculative --- what is the role of high number of rainy days? ..indirectly favouring high air humidity? else?

Page 12,

Lines 250-253

Other (more appropriate) studies that have looked at the difference in predictive skill of crop simulation models for winter and spring wheat include the following:

- Pirttioja, N., Carter, T.R., Fronzek, S., Bindi M, Hoffmann H, Palosuo T, Ruiz-Ramos M, Tao F, Trnka M, Acutis M, Asseng S, Baranowski P, Basso B, Bodin P, Buis S, Cammarano D, Deligios P, Destain MF, Dumont B, Ewert F, Ferrise R, François L, Gaiser T, Hlavinka P, Jacquemin I, Kersebaum KC, Kollas C, Krzyszczak J, Lorite IJ, Minet J, Minguez MI, Montesino M, Moriondo M, Müller C, Nendel C, Öztürk I, Perego A, Rodríguez A, Ruane AC, Ruget F, Sanna M, Semenov MA, Slawinski C, Stratonovitch P, Supit I, Waha K, Wang E, Wu L, Zhao Z and Rötter, R.P. (2015). A crop model ensemble analysis of temperature and precipitation effects on wheat yield across a European transect using impact response surfaces. *Climate Research*, 65: 87–105. doi:10.3354/cr01322

- Ruiz-Ramos, M., R. Ferrise, A. Rodríguez, I. J. Lorite, M. Bindi, T.R. Carter, S. Fronzek, T. Palosuo, N. Pirttioja, P. Baranowski, S. Buis, D. Cammarano, Y. 4 Chen, B. Dumont, F. Ewert, T. Gaiser, P. Hlavinka, H. Hoffmann, J.G. Höhn, F. Jurecka, K.C. Kersebaum, J. Krzyszczak, M. Lana, A. Mechiche-Alami, J. Minet, M. Montesino, C. Nendel, J.R. Porter, F. Ruget, M. A. Semenov, Z. Steinmetz, P. Stratonovitch, I. Supit, F. Tao, M. Trnka, A. de Wit and R. P. Rötter (2017). Adaptation response surfaces for managing wheat under perturbed climate and CO₂ in a Mediterranean environment. *Agricultural Systems* DOI: 10.1016/j.agsy.2017.01.009

Lines 257-258 – please, correct – it is not true that heat stress only impacts wheat during the short flowering period --- also post flowering heat can have detrimental effects on yield due to accelerated leaf senescence (see Assen et al 2011, in *Global Change Biology*).

Pages 14 to 16

I miss a short discussion on the possible effects of elevated CO₂ in the future respectively its interaction with the other climate variables – please, consider to add a little paragraph

Supplementary Information

- The legend of figure S1 a is confusing – please, clarify

Point-by-point answer to comments on NCOMMS-17-18455A

Please find our point-by-point answers to the comments provided by two referees on our manuscript entitled «Causes and implications of the unforeseen 2016 extreme yield loss in France's breadbasket».

The comments themselves are in italic and our answers as normal text. Required changes have been highlighted in bright yellow in the resubmitted manuscript.

Additionally to the changes strictly required by the two referees, we decided to also include a few additional changes in the manuscript. All changes can be found as track changes in the manuscript. The most important change is that (i) we replaced "Europe" with "France" in the title and that, (ii) we decided to move all results referring to *net* yield losses (i.e., yield anomalies below 0) to the supplement. Our results are consistent across yield losses, but we believe that this can alleviate the presentation of the results. Focusing on severe and extreme losses only also seems more consistent with the description of the 2016 extreme event. We do hope that the two will referee agree with these additional changes.

Referee #1: COMMENTS TO THE AUTHOR (S)

I found this study to be very interesting. It takes a recent event that is puzzling, asks how unusual it was in the historical context, and develops an understanding of its causes and spells out implications for the future. I learned quite a bit from it. I think it should be published, but I have a few questions/suggestions.

My main question is about the model itself. The authors make an unconventional choice in modeling the occurrence of loss above certain amounts (i.e. a binary variable) rather than modeling the yield itself. This raises the question of whether conclusions would be much different if modeling yield itself, or %yield anomaly. I would suggest the authors conduct this alternative analysis and present the results. They could either revisit variable selection for this new outcome variable, and/or could test predictions if using the same predictors but a different outcome. This would help strengthen the paper and convince the reader the exact specification isn't too important (or if it turns out it is, the authors should explain why their approach makes more sense).

The objective of our work is to identify plausible risk factors associated with high levels of yield loss, but not to explain yield variability over the whole range of yield values. The chosen modelling approach (i.e., binomial logistic regression) is commonly used in different research fields for selecting risk factors and estimating probabilities of occurrence of events. This approach was thus well adapted to our objective.

However, for the sake of comparison, we conducted the alternative analysis suggested by the reviewer. We fitted linear models to normalized anomalies of yields, and applied the same variable selection procedure based on BIC (i.e., univariate selection and then multivariate selection without and with interactions). We found that the model leading to the lowest BIC included the same variables as those of the binomial logistic model, i.e.,

X1 is the number of days between 0 and 10°C in the fall, X2 is the number of rainy days in November, X5 is minimum temperature in June and X6 precipitation in the spring (AMJJ), i.e., the 4 variables with lowest individual BIC presented in (i). The results of these additional analyses are presented in the response letter only.

Note that two interactions are also selected – just as with the binary data, namely the interaction between temperatures in the fall and precipitation in the spring and the interaction between temperature in June and precipitation in the spring. All covariates are highly significant, as shown below:

We show that:

(i) Using models based on normalized yield anomalies leads to selecting the same variables (see below for the 10 variables with lowest BIC in each season).

# FALL	BIC	# SPRING	BIC
tx_0_10_D	-2632.488818	tn_Ju	-2766.416171
pr_per__N	-2615.456669	pr_AMJJ	-2633.937061
pr_N	-2601.329462	tn_AMJJ	-2632.056773
pr_per__OND	-2587.890309	tx34_Ju	-2622.099476
pr_OND	-2586.622338	pr_Ju	-2609.136532
tx_0_10_OND	-2583.248622	pr_per__AMJJ	-2598.025177
tx_O	-2580.00158	pr_per__A	-2595.455717
tn_D	-2578.307003	pr_per__Ju	-2594.071616
tn_N	-2577.13018	tx_Ju	-2590.628496
etp_D	-2576.402856	tx34_AMJJ	-2588.224756

(ii) Combining these variables and applying step BIB leads to the selection of the same model as the one based on the occurrence of yield losses. See model R summary below.

Call:
glm(formula = TAB\$anomaly_norm ~ X1 + X2 + X5 + X6 + X1:X6 +
X5:X6)

Deviance Residuals:

Min	1Q	Median	3Q	Max
-0.38194	-0.05886	0.00266	0.06118	0.25679

Coefficients:

	Estimate	Std. Error	t value	Pr(> t)
(Intercept)	1.331413	0.100866	13.200	< 2e-16 ***
X1	-0.007853	0.002134	-3.679	0.000242 ***
X2	-0.079007	0.013688	-5.772	9.43e-09 ***
X5	-0.085562	0.007764	-11.020	< 2e-16 ***
X6	-0.565669	0.053155	-10.642	< 2e-16 ***
X1:X6	0.006603	0.001013	6.517	9.69e-11 ***
X5:X6	0.031291	0.003947	7.928	4.20e-15 ***

Signif. codes: 0 '***' 0.001 '**' 0.01 '*' 0.05 '.' 0.1 ' ' 1

(Dispersion parameter for gaussian family taken to be 0.008318846)

Null deviance: 17.653 on 1565 degrees of freedom
Residual deviance: 12.969 on 1559 degrees of freedom
AIC: -3046.8

Number of Fisher Scoring iterations: 2

(iii) Predictions of normalized yield anomalies and predicted probabilities of yield loss follow similar patterns (as we show in Figure 1). Notably, major loss events occur during the same years (e.g., 1976, 2003 and 2016) with both types of model. A few additional moderate loss years were also identified by the linear model (1966,1970 and 2007).

Figure 1. Outputs of a linear model fitted to normalized yield anomalies. Input variables were selected using a BIC stepwise procedure (as for the binomial logistic regressions). The selected inputs are: the number of cold days in december, precipitation in November, temperature in June and precipitation in the spring (see Table S2).

The paragraph starting on L49 could use some more details on the yield loss, particularly what was associated with it in terms of diseases. This becomes an important part of the story but is only really clarified on L347 that there was a specific disease that was thought to cause much of the loss.

We have added a few sentences (L51-61) reflecting the contents of technical documents produced by regional experts. Note that an important biomass on the one hand and the widespread use of fungicides on the other, possibly explains why most experts assumed average yields until harvest started unfolding in mid-July.

L155-156. Not clear why the second sentence says from 0.61 and not from 0.66

This sentence was unclear. We changed the sentence and now present the results for extreme losses (we moved all results referring to net yield losses to the supplement and changed Figure 4 to present severe and extreme losses probabilities). See L164-171.

Figure legends and axes are hard to read. Please make font bigger when possible

We increased legend sizes on all plots (main and supplement).

My understanding was that severe losses are anything > 10% whereas extreme

is >15%. Therefore severe has to be more likely than extreme, by definition. Yet on L233-34 it says "Estimated probabilities of net, severe, and extreme yield loss in 2016 are on average of 0.71, 0.46 and 0.8 respectively"

The reason is that the selected set of inputs was different for severe and extreme yield losses, leading to two separate models. The former includes one additional fall*spring interaction (see Table S2). The advantage of this strategy is to allow us selecting the best set of input variables for each yield loss threshold (the best set of inputs can depend on the chosen threshold). But, as a consequence, the probability of extreme yield loss is not constrained to be systematically lower than the probability of severe yield loss. We added this information in the text (see for example L414).

Furthermore, if we refer to table S3 (copied below), which summarizes the values of estimated probabilities in 2016 including or excluding 2016 in the training dataset, it shows that the gap between prior and posterior probabilities for net and severe yield losses are in fact very close to each other (i.e., 0.32-0.34). Note that we changed the between-department range from the minimum and maximum probabilities that was displayed previously to the 10th and 90th percentiles.

Table S3. Odds and risk ratios of net, severe and extreme yield loss in the French Breadbasket in 2016 computed from prior and estimated probabilities from a dataset trained in the study area from 1958 to 2016 (dataset including the 2015-2016 growing season) and from 1958 to 2015 (dataset excluding the 2015-2016 growing season).

NET YIELD LOSS					
Median values across counties					
	Prior	Posterior	Probability	Risk ratio	Odds ratio
Training dataset Excl. 2016	0.46	0.80	(0.74-0.84)	1.74	4.80
Training dataset Incl. 2016	0.47	0.85	(0.81-0.89)	1.81	6.57

SEVERE YIELD LOSS					
Median values across counties					
	Prior	Posterior	Probability	Risk ratio	Odds ratio
Training dataset Excl. 2016	0.14	0.46	(0.32-0.56)	3.23	5.11
Training dataset Incl. 2016	0.16	0.69	(0.49-0.84)	4.42	12.03

EXTREME YIELD LOSS					
Median values across counties					
	Prior	Posterior	Probability	Risk ratio	Odds ratio
Training dataset Excl. 2016	0.07	0.71	(0.23-0.97)	10.67	34.83
Training dataset Incl. 2016	0.08	0.93	(0.42-0.99)	11.14	136.37

L382 (reference to wrong table)

Thank you for noticing this (there was another false reference to Table S1 below that we also corrected). We now refer to Table S2.

I'd suggest a simple figure at the end showing the probability of extreme yield loss over time for rcp2.6 and 8.5. this would aid discussion already in the text, and provide a nice summary visual for the main point. If space is needed, fig 5 could be moved to supplement.

We believe that there are at least two important impediments. First, the spatial resolutions of the climate projection used (a rectangle with latitude and longitude values matching the study area) and of the yield data (administrative polygons) are not exactly the same. Second, making robust quantitative statement on the evolution of the probability of extreme yield loss in the future perhaps requires a dedicated study. Indeed, Factors that are not necessarily crucial in the present climate may become much more important when dealing with the end of the 21st century (such as the changes in the CO₂ concentration). Additionally, as we are dealing with extreme yield losses, it would likely be necessary to remove the bias from the entire statistical distribution of climate variables and take into account the co-variability of climate variables in the bias correction algorithm. Uncertainty on spring precipitation - one of the major challenge in climate projection exercises so far - also possibly hampers our ability to accurately predict future changes in wheat yield loss probability resulting from a 2016-year like event.

If the referee agrees, we added a sentence referring explicitly to this matter in the discussion (see L 352-359).

The supplemental figures were not numbered nor did they have captions in the version I have, which is annoying.

Done. The reason for that is that the figure number was specified in the files names, thus not apparent in the merged document. We changed this and added numbers in the supplement figures.

Referee #2: COMMENTS TO THE AUTHOR (S)

GENERAL COMMENTS

The manuscript NCOMMS-17-18455 by Ben-Ari et al., entitled "Causes and implications of the unforeseen 2016 extreme yield loss in Europe's breadbasket" tackles a very important, hot research Topic, i.e. crop yield impacts of extremes, and addresses some fundamental research questions, such as: (1) how exceptional were the climate conditions, individually and combined (2) why did the existing yield forecasting Systems fail to anticipated the extreme yield loss? (3) how can - on the basis of this experience and detailed analyses - wheat yield forecasting systems be improved in the future. (4) will the observed combined events become more frequent in the future? The research presented is original, yields several novel results and may have considerable influence on future directions of crop modelling /yield forecasting. The paper presents a sound combination of knowledge and methods in the fields of climate projections, crop modelling and yield forecasting. Moreover, the paper is very well written, contains key references and uses adequate figures (very nice illustrating key results) with very few exceptions (see, specific comments). Very minor points found that ask for amendment (see, specific comments)

Overall judgment: minor revision

SPECIFIC COMMENTS

Page 2, line 51: you may want to mention the total production achieved in 2015

This is now added in the text (Line 47-49) and indeed helps getting an idea on the relative size of the production loss.

Line 53-55 would be interesting to know how much the low exchange prices contributed to income loss in comparison to the yield reduction effect

We relied on the observatory of economic complexity database to extract more precise information on the loss of 2016 (<https://atlas.media.mit.edu/en/>). We find that the gap between wheat export values in 2016 versus the average from 2010 to 2015 is about 2.3 billion dollars (L52).

Page 6, Lines 117-119: please, add a brief explanation on why you choose the values below 0, -10 and -15% for net, severe and extreme ? – some reference to earlier /other work?

This was clarified. The threshold of -10% corresponds to a loss of about one standard deviation. The -15% threshold corresponds to the 10th percentile of the yield anomalies. We chose the names severe and extreme to simplify the text. Note that we also decided to move all results corresponding to a loss below 0 (net loss) to the supplement to focus on -10 and -15% losses and to alleviate the text; we hope the reviewer agrees with this change. Figure 4 is now a panel for severe and extreme interactions and Figure S6 a panel for net yield losses for both interactions. See new paragraph L122-127.

Lines 121-122 maybe you could explain why you did categorically not consider climate variables of months January, February and March?

Done. The variable selection procedure relies both on a description of the extreme weather conditions of the 2015-2016 growing season and on statistical tests. Our objective was to test if the effect of the compound weather extreme described for the 2015-2016 growing season was significantly associated with yield losses when taking into account the totality of the yield time series. The winter of 2016 was relatively warm and rainy, but did not show any extremes (see Fig.2). On the contrary, fall and spring showed very extreme weather conditions, and we thus focused on these two seasons. We tried to make the selection procedure more explicit in the revised manuscript (see for example, L131-133).

Based on the reviewer's comment, we decided to assess the robustness of our conclusions to the inclusion of winter weather variables. We re-computed all results including winter variables for winter (JFM temperatures and precipitations, See Table attached to this answer). We found that all the variables previously selected (i.e., fall and spring conditions and their interactions) were still selected by our statistical procedure when winter variables were included. Importantly, the interactions that we describe in the article remain unchanged (see figure below).

Figure 2. Probability of severe yield losses. Interaction between the number of days between 0 and 10°C in December and spring precipitation (left) for three levels of spring precipitation and interaction between temperature in June and precipitation in the spring (right). These results were obtained with a model including winter weather variables. This figure is presented in the response letter only.

Line s 128-129 could you say a bit more (than summary in Table S1) about the composition of expert panels and the method of expert interviews? – would be interesting to learn more about this.

There was confusion in the text here. In fact this sentence refers to information that we extracted from documents produced by the academy of agriculture, technical institutes or specialized newspapers (i.e., grey literature expertise). We slightly changed the sentence here to alleviate any confusion (see L138-140).

Page 9, Lines 174-178 I would suggest to also refer here (or later in the discussion) to a related paper on the expected higher frequency of adverse weather on wheat production in Europe -> Trnka, M., Rötter, R.P., Ruiz-Ramos, M. Kersebaum, K-C., Olesen, J.E. & Semenov, M.A. (2014). Adverse weather conditions for wheat production in Europe will become more frequent with climate change. Nature Climate Change, 4, 637-643 (doi: 10.1038/nclimate2242).

Done. We now refer to the article from Trnka et al, on the frequency of combined adverse events over Europe. See L184.

Lines 174-178 inform how the lack of sufficient vernalizing days can modify subsequent development

We added two sentence and two references on page 9 L199-202 on this topic, and also added a short sentence earlier in the manuscript on L101-103.

Lines 192-193 – would be good if you could find quantitative information on how much durum wheat was affected (and possibly a reference?)

The average loss for durum wheat in France was -20% in 2016 compared to the 5 years average, according to the national database for crop yield AGRESTE (which we now cite, L205-207).

Lines 197 to 201 (page 10) – is there some literature already that supports your hypotheses? - Please, check for such and add here

Done. We added one sentence on this topic. See L21-215, p10. In this article by J. West et al, it is said that: *“Research to predict effects of climate change on the wheat disease, septoria leaf spot in France, concluded that predictions were difficult due to contradictory effects of mild weather promoting inoculum build-up over winter but drier weather reducing infection of final leaves in late-spring(Gouache et al.2011). The early stages of disease are likely to be enhanced because the intercrop survival of the pathogen is favored by dry summers, (Shaw et al.2008) and because infection success (of spores) is promoted by milder winter weather (>7°C) (Pietravalle et al. 2003; Beest et al. 2009). It seems clear that these factors are likely to lead to an increase in this disease on leaves at the base of the plant over winter and early spring. However, Gouache et al (2011) concluded that in France, despite some advancement of wheat phenology, declining spring rainfall events will reduce spread of this disease onto the final leaf layers. It should be noted that in the event of a wet spring, there may be capacity for this disease to be more severe than would be expected currently.”*

Page 10, Lines 202-210 this section, though interesting, does not say much about the exact mechanisms leading to yield loss –specify the prevailing processes and their effects, please

We agree. This paragraph attempt to do two things ; (i) provide plausible mechanisms (i.e., these are not tested here due to a lack of specific data) and (ii) discuss them in the light of observed weather in 2016. We tried to clarify the totality of the paragraph (starting from “underlying mechanisms”) by re-writing large sections of it. We added a few references, clarified the section on vernalization and removed some unclear sentences.

Lines 211-215 – this becomes now a bit too speculative --- what is the role of high number of rainy days? .. indirectly favouring high air humidity ? else ?

This is true. We removed the most speculative section of this small paragraph (L232-235).

Page 12, Lines 250-253: Other (more appropriate) studies that have looked at the difference in predictive skill of crop simulation models for winter and spring wheat include the following:

We agree. Two references were added:

*- Pirttioja, N., Carter, T.R., Fronzek, S., Bindi M, Hoffmann H, Palosuo T, Ruiz-Ramos M, Tao F, Trnka M, Acutis M, Asseng S, Baranowski P, Basso B, Bodin P, Buis S, Cammarano D, Deligios P, Destain MF, Dumont B, Ewert F, Ferrise R, François L, Gaiser T, Hlavinka P, Jacquemin I, Kersebaum KC, Kollas C, Krzyszczak J, Lorite IJ, Minet J, Minguez MI, Montesino M, Moriondo M, Müller C, Nendel C, Öztürk I, Perego A, Rodríguez A, Ruane AC, Ruget F, Sanna M, Semenov MA, Slawinski C, Stratonovitch P, Supit I, Waha K, Wang E, Wu L, Zhao Z and Rötter, R.P. (2015). A crop model ensemble analysis of temperature and precipitation effects on wheat yield across a European transect using impact response surfaces. *Climate Research*, . 65: 87–105. doi:10.3354/cr01322*

*- Ruiz-Ramos, M., R. Ferrise, A. Rodríguez, I. J. Lorite, M. Bindi, T.R. Carter, S. Fronzek, T. Palosuo, N. Pirttioja, P. Baranowski, S. Buis, D. Cammarano, Y. 4 Chen, B. Dumont, F. Ewert, T. Gaiser, P. Hlavinka, H. Hoffmann, J.G. Höhn, F. Jurecka, K.C. Kersebaum, J. Krzyszczak, M. Lana, A. Mechiche-Alami, J. Minet, M. Montesino, C. Nendel, J.R. Porter, F. Ruget, M. A. Semenov, Z. Steinmetz, P. Stratonovitch, I. Supit, F. Tao, M. Trnka, A. de Wit and R. P. Rötter (2017). Adaptation response surfaces for managing wheat under perturbed climate and CO2 in a Mediterranean environment. *Agricultural Systems* DOI: 10.1016/j.agry.2017.01.009*

*Lines 257-258 – please, correct – it is not true that heat stress only impacts wheat during the short flowering period --- also post flowering heat can have detrimental effects on yield due to accelerated leaf senescence (see Assen et al 2011, in *Global Change Biology*).*

We changed this sentence and now cite the Asseng (2011) on temperature variability here (L276-277).

Pages 14 to 16: I miss a short discussion on the possible effects of elevated CO₂ in the future respectively its interaction with the other climate variables – please, consider to add a little paragraph.

We inserted the following paragraph (L346-351):

Note that we did not address the possible effects on wheat yields of an increase in atmospheric CO₂ concentration. CO₂ effects are expected to manifest through increased leaf level carboxylation rates and stomatal closure. Both processes interact with each other, and stomatal closure has the dual consequence of saving soil water and increasing surface temperature by reducing transpiration. It is thus expected that elevated CO₂ will improve yields be under dry condition (53, 54).

Supplementary Information: - The legend of figure S1 a is confusing – please, clarify

We modified the legends in Figure S1.

REVIEWERS' COMMENTS:

Reviewer #1 (Remarks to the Author):

this is a good paper and improved from its original version. i think it should be published with a couple minor revisions

L351 has a typo

the response to question 1 by reviewer 1 was fairly thorough, but none of it was mentioned in the actual paper. it seems like they could briefly mention the similar results somewhere on page 6 or 7

Reviewer #2 (Remarks to the Author):

I am very pleased with the thorough, to the point replies to my comments by the authors; I also acknowledge the careful replies of the authors to reviewer #1 comments. Generally, by further adjustments made by the authors the paper now reads even better than the earlier version and is easier to comprehend for a broader public.

I accept as is - just found one typo (atmospheric - corrected below) and a "remnant word" in last sentence of this paragraph ("be" that preceded "under dry condition - deleted now):

(L346-351):

Note that we did not address the possible effects on wheat yields of an increase in atmospheric CO₂ concentration. CO₂ effects are expected to manifest through increased leaf level carboxylation rates and stomatal closure. Both processes interact with each other, and stomatal closure has the dual consequence of saving soil water and increasing surface temperature by reducing transpiration. It is thus expected that elevated CO₂ will improve yields under dry condition (53, 54).

s

Point-by-point answer to comments on NCOMMS-17-18455A

Referee #1: COMMENTS TO THE AUTHOR (S)

This is a good paper and improved from its original version. I think it should be published with a couple minor revisions

L351 has a typo

The type ('be' misplaced) has been corrected

The response to question 1 by reviewer 1 was fairly thorough, but none of it was mentioned in the actual paper. it seems like they could briefly mention the similar results somewhere on page 6 or 7

The editors proposed that we include this section into the supplement. We now provide a small paragraph on this particular point in the Supplementary Discussion section.

Referee #2: COMMENTS TO THE AUTHOR (S)

I am very pleased with the thorough, to the point replies to my comments by the authors; I also acknowledge the careful replies of the authors to reviewer #1 comments. Generally, by further adjustments made by the authors the paper now reads even better than the earlier version and is easier to comprehend for a broader public.

I accept as is - just found one typo (atmospheric - corrected below) and a "remnant word" in last sentence of this paragraph ("be" that preceded "under dry condition - deleted now):

Yes, this was also noticed by Ref1. The word was removed and missing letter added in the former:

(L346-351):

Note that we did not address the possible effects on wheat yields of an increase in atmospheric CO₂ concentration. CO₂ effects are expected to manifest through increased leaf level carboxylation rates and stomatal closure. Both processes interact with each other, and stomatal closure has the dual consequence of saving soil water and increasing surface temperature by reducing transpiration. It is thus expected that elevated CO₂ will improve yields under dry condition (53, 54).

Missing s was added.